# Lichenysin-like Polypeptide Production by *Bacillus licheniformis* B3-15 and Its Antiadhesive and Antibiofilm Properties

**DOI:** 10.3390/microorganisms11071842

**Published:** 2023-07-20

**Authors:** Vincenzo Zammuto, Maria Giovanna Rizzo, Claudia De Pasquale, Guido Ferlazzo, Maria Teresa Caccamo, Salvatore Magazù, Salvatore Pietro Paolo Guglielmino, Concetta Gugliandolo

**Affiliations:** 1Department of Chemical, Biological, Pharmaceutical and Environmental Sciences, University of Messina, Viale Ferdinando Stagno D’Alcontres 31, 98166 Messina, Italy; vzammuto@unime.it (V.Z.); sguglielm@unime.it (S.P.P.G.); 2Research Centre for Extreme Environments and Extremophiles, Department of Chemical, Biological, Pharmaceutical and Environmental Sciences, University of Messina, Viale Ferdinando Stagno D’Alcontres 31, 98166 Messina, Italy; mariateresa.caccamo@unime.it (M.T.C.); smagazu@unime.it (S.M.); 3ATHENA Green Solutions S.r.l., Viale Ferdinando Stagno d’Alcontres 31, 98166 Messina, Italy; 4Laboratory of Immunology and Biotherapy, Department of Human Pathology, Via Consolare Valeria, 1, 98124 Messina, Italy; cdepasquale@unime.it; 5Department of Experimental Medicine (DIMES), University of Genoa, 16132 Genova, Italy; guido.ferlazzo@unige.it; 6Unit of Experimental Pathology and Immunology, IRCCS Ospedale Policlinico San Martino, 16132 Genova, Italy; 7Department of Mathematical and Computer Sciences, Physical Sciences and Earth Sciences, University of Messina, Viale Ferdinando Stagno D’Alcontres 31, 98166 Messina, Italy

**Keywords:** antiadhesive, antibiofilm, *Bacillus*, biosurfactant, bioemulsifier, human cell viability

## Abstract

We report the ability of the crude biosurfactant (BS B3-15), produced by the marine, thermotolerant *Bacillus licheniformis* B3-15, to hinder the adhesion and biofilm formation of *Pseudomonas aeruginosa* ATCC 27853 and *Staphylococcus aureus* ATCC 29213 to polystyrene and human cells. First, we attempted to increase the BS yield, optimizing the culture conditions, and evaluated the surface-active properties of cell-free supernatants. Under phosphate deprivation (0.06 mM) and 5% saccharose, the yield of BS (1.5 g/L) increased by 37%, which could be explained by the earlier (12 h) increase in *lch*AA expression compared to the non-optimized condition (48 h). Without exerting any anti-bacterial activity, BS (300 µg/mL) prevented the adhesion of *P. aeruginosa* and *S. aureus* to polystyrene (47% and 36%, respectively) and disrupted the preformed biofilms, being more efficient against *S. aureus* (47%) than *P. aeruginosa* (26%). When added to human cells, the BS reduced the adhesion of *P. aeruginosa* and *S. aureus* (10× and 100,000× CFU/mL, respectively) without altering the epithelial cells’ viability. As it is not cytotoxic, BS B3-15 could be useful to prevent or remove bacterial biofilms in several medical and non-medical applications.

## 1. Introduction

Surface-active molecules (SAMs) are produced by a large variety of microorganisms to increase the bioavailability of hydrophobic compounds as potential carbon and energy sources [1]. Bacterial SAMs can be distinguished into two different classes according to their molecular weight: biosurfactants (BSs) and bioemulsifiers (BEs). BSs, including lipopeptides, glycolipids, and phospholipids, are compounds characterized by low molecular weight and efficiently reduce both surface and interfacial tension [2,3,4]. Unlike BSs, BEs are able to efficiently emulsify hydrophobic substrates, and they have higher molecular weight, since they are constituted by a mixture of biopolymers (such as polysaccharides, lipopolysaccharides, lipoproteins, and proteins) [4,5,6]. Due to the marked differences between biosurfactants and bioemulsifiers, their roles in nature and biotechnological applications are different [4,5].

Different microorganisms produce lipopeptide biosurfactants, which are secondary metabolites with important applications in the oil industry and the agricultural and medical fields. Based on their structural components (i.e., amino acid chain and fatty acids), cyclic lipopeptides produced by *Bacillus* species have in five distinct families, which exhibit notable surface activity with important biological effects [7,8]. Iturin (produced by *Bacillus subtilis*, *B. amyloliquefaciens*, *B. licheniformis*, *B. thuringiensis* and *B. methyltrophicus*) [9,10,11,12,13,14,15,16,17], fengycin (*B. cereus* [18] and *B. thuringiensis* [19], in addition to *B. subtilis* [20] and *B. amyloliquefaciens* [21]), kurstakin (*B. thuringiensis* [22]), locillomycin (*B. subtilis* 916 [23]), and surfactin, including pumilacidin and lichenysin (isolated from *B. coagulans* [24], *B. pumilus* and *B. licheniformis* [25]), are well-recognized antimicrobial compounds that are used in biotechnological and biopharmaceutical applications [26].

The lipopeptide lichenysin differs from surfactin in terms of a change in the first amino acid residue, glutamine (Gln), instead of glutamic acid. Based on the molecular weight and structures, the lichenysins have been distinguished into different types (A, B, C, D, and G and the surfactant BL86), and they are reported to have similar surface-active properties, with good solubilizing, foaming, emulsifying, and detergent activity, which are useful in a broad range of applications [27,28,29,30]. Lichenysin production is frequently observed in *Bacillus licheniformis*, a nonpathogenic bacterium that is unable to penetrate the skin or mucous membranes of the body in the absence of pre-existing lesions. The lichenysin peptide biosynthesis is carried out by non-ribosomal peptide synthetases, and proteins are encoded by the *lch*AA-AB-AC-TE gene cluster, also annotated as *lic*A-TE [31]. The presence of the *lch*AA gene in *B. licheniformis* appears to be very common since it was detected in all the strains studied by Madslien et al. [25]. However, lichenysin production depends on the level of transcription and enzyme (lichenysin synthetase) activity [32], and it is strongly influenced by culture conditions, such as the type of carbon, nitrogen, or phosphate sources [33].

Other than exhibiting several bioactive properties (antitumor, antimicrobial, and antioxidant properties) ([34] and references therein], lichenysin was found to possess noteworthy antiadhesion activity, being able to prevent the formation of and eradicate the bacterial biofilm [27,33]. Bacterial biofilms on abiotic and biotic surfaces are important in different fields, such as food spoilage, biofouling, and human health [34,35,36,37,38,39], principally due to the high tolerance and persistence of biofilms with respect to disinfectants and antimicrobial agents [37,40]. Biofilm formation is a dynamic and complex process, starting with the adhesion of bacterial cells to surfaces and evolving into biofilm maturation and cell dispersion in order to contaminate other surfaces [4,41]. To contrast with the development of bacterial biofilm, it could be useful to know the phases involved in its formation. The first stage of bacterial attachment to surfaces is considered a critical step for the successful formation of biofilms, since environmental conditions (temperature, pH, salinity, etc.), surface properties (substrate type, surface roughness, and chemical composition), cell-surfaces charges, and hydrophobicity may affect the establishment of biofilms [42,43,44].

The inhibition of bacterial adhesion and biofilm formation represents a pivotal approach to contrast with diseases related to persistent microbial infection. Previously, we have reported that *B. licheniformis* B3-15 was able to produce a surfactin-like lipopeptide (BS B3-15), with specific activity in emulsifying and removing hydrocarbons and vegetable oils [45]. Due to its properties, this biosurfactant was suggested to be an environmentally friendly compound that is useful in different fields of applications.

The aim of this work was to investigate the effects of BS B3-15 addition on the first adhesion and biofilm formation of *Pseudomonas aeruginosa* ATCC 27853 and *Staphylococcus aureus* ATCC 29213 on a polystyrene surface and in human nasal epithelial cells. First, we attempted to enhance the BS B3-15 production yield, optimizing the culture medium, using the Response Surface Methodology (RSM). Biosurfactant production was verified using both the surface-active properties of the cell-free supernatant and the expression of the lichenysin synthetase A gene (*lch*AA) at different times of bacterial growth (12 h, 24 h, and 48 h). To evaluate the effects of *P. aeruginosa* and *S. aureus* on biofilm formation, crude BS B3-15 was added to polystyrene surfaces at different times, corresponding to the different phases of biofilm formation (i.e., initial attachment, reversible attachment, irreversible attachment) and after the biofilm was established. In order to address the BS B3-15 use in different medical and non-medical applications, the effects of the crude BS were evaluated on the adhesion of *P. aeruginosa* and *S. aureus* to human nasal epithelial cells, constituting the first targets of air-mediated bacterial infections.

## 2. Materials and Methods

### 2.1. Bacillus licheniformis Strain B3-15

The *Bacillus licheniformis* strain B3-15 has been previously reported as capable of producing BS B3-15 [45]. Briefly, strain B3-15 was isolated from thermal water emitted from a shallow hydrothermal vent near the Vulcano-Island Porto di Levante (Aeolian Islands, Italy), characterized by a depth of 5 m, and from temperature and pH water of 65 °C and 5.2, respectively [45]. The strain grew aerobically, with the lower-limit temperature of 25 °C and the upper-limit temperature of 60 °C. The pH range for growth was 5.5–9, with an optimal value of 7, and the strain grew in NaCl concentration ranging from 0 to 7% (*w*/*v*), with the optimum occurring at 2% (*w*/*v*). Furthermore, a partial 16S rRNA gene of the strain was sequenced, and the following accession number was assigned KC485000. To maintain the strain, B3-15 Tryptic Soy Agar (TSA, Sigma Aldrich, Burlington, MA, USA) plates, supplemented with 1% NaCl (1.5% NaCl final concentration), were used. The BS B3-15 was produced in Marine Broth supplemented with 3% of saccharose, and, after acid precipitation, its yield was 910 mg/L. The BS B3-15 was spectroscopically characterized as a surfactin-like lipopeptide.

### 2.2. Optimization of Culture Conditions for BS B3-15 Production

To evaluate the effects of the saccharose or glucose and phosphate content on the strain growth and BS production, the strain B3-15 (OD_600nm_ = 0.1, corresponding to 1.7 × 10^8^ CFU/mL, as experimentally evaluated) was inoculated in the novel medium, named MGV, containing different concentrations of saccharose or glucose (0, 3, 5, 7% *w*/*v*), and K_2_HPO_4_ (0.008–8 g/L) and KH_2_PO_4_ (0.002–2 g/L) and incubated at optimal growth temperature of 45 °C for 48 h on rotating shakers (250 rpm) (Table 1).

The cell-free supernatant (CFS) was obtained using centrifugation at 8000 rpm for 10 min and filtration through a 0.2 µm pore size membrane (Biogenerica, Catania, Italy) of the culture. Aliquot (100 µL) of the filtrated supernatant was placed onto plates of TSA and incubated at a temperature of 45 °C for 24 h to verify that no bacterial cells were present. The CFSs were used for surface property assays.

#### 2.2.1. Surface-Active Properties of Cell-Free Supernatant (CFS)

The CFS’s surface-active properties were evaluated through the oil’s drop-collapse and emulsification assays. To perform the oil’s drop-collapse assay, a drop (100 µL) of each CFS was spotted on the polystyrene lid of a 96-microwell plate (Thermo-Fisher, Milan, Italy), and then mineral oil (5 µL) (Sigma Aldrich) was added. The CFS containing BS gave a flat drop.

The emulsification assay of each CFS was carried out as described by Tuleva et al. [46]. Briefly, each CFS or uninoculated sterile medium, used as negative controls, was mixed in a glass tube (10 cm high and 1 cm in diameter) with an equal volume of kerosene (Sigma Aldrich), vortexed vigorously, and left to stand for 24 h. The tubes were incubated at room temperature for 24 h, and the emulsion index was calculated according to the following formula:$$E24=\frac{High\;of\;emulsion}{High\;of\;the\;whole\;solution}\times 100$$

#### 2.2.2. Response Surface Methodology

The effect of the carbohydrate content (0–7%) and the phosphate (K_2_HPO_4_ and KH_2_PO_4_) concentration (from 0.06 to 60 mM) in the phosphate buffer on the overall biosurfactant production (evaluated by E_24_ of CFS) was analyzed in a set of 13 experiments using the Response Surface Methodology—the Central Composite Design (RSM-CCD) tool of Minitab (version 21.0) statistical software. The interaction was expressed as the quadratic model equation to predict the optimized composition medium named MGV-op.

### 2.3. BS B3-15 Extraction, Characterization, and Surface Activity

#### 2.3.1. BS B3-15 Extraction

To extract the BS, each CFS was acidified at pH 2.0 using 2N HCl and kept overnight to allow precipitation [42,47]. An equal volume of a chloroform: methanol (2: 1 *v*/*v*) mixture was added to the acidified CFS, and the organic layer (extract) was separated from the aqueous phase (solvent). The BS dissolved in the organic phase was recovered using a rotary vapor treatment (Rotavapor^®^ R-300, BUCHI Italia S.r.l, Cornaredo, Italy) and a desiccation process (45 °C, overnight). The obtained BSs were finally weighed and stored at 4 °C.

#### 2.3.2. Characterization of BS B3-15 using ATR-FTIR

To identify the functional groups of the crude BS, which was obtained under optimized conditions, the attenuated total reflectance Fourier Transform Infrared (ATR-FTIR) technique was used. VERTEX 70v FT-IR Spectrometer (Bruker Optics GmbH & Co. KG, Ettlingen, Germany) equipped with the platinum diamond was employed to obtain the spectra in the wavenumber range of 4000 to 400 cm^−1^ with a resolution of 4 cm^−1^. Finally, OMNIC software version 7.3 (Origin Lab Co., Northampton, MA, USA) was employed to analyze the peaks of the BS spectrum.

#### 2.3.3. BS B3-15 Surface-Active Properties

To assess the ability of the crude BS B3-15 to modify the hydrophobic surfaces at increasing concentrations (from 0 to 1600 µg/mL), the contact angle (θ) in the water solution was measured using the sessile-drop technique. Briefly, 5 µL of each BS B3-15 solution were spotted onto a lid of a 96-well microtiter plate (Thermo Fisher Scientific, Waltham, MA, USA), incubated at room temperature for 15 min, and photographed with a high-resolution camera, as previously reported [48]. To measure the angle (θ) on the sessile drop images, the images were analyzed (in triplicate) using the software ImageJ 1.54d (ImageJ, National Institutes of Health, Bethesda, MD, USA). Drop Snake plugin, and the average value and standard deviation were calculated.

### 2.4. Expression of Lichenysin and Surfactin Synthetase

#### 2.4.1. Gene Expression Analysis

Genes related to the production of lichenysin and surfactin, detected during the growth of *B. licheniformis* B3-15 in MGV and MGV-op, are reported in Table 2.

To determine the primer characteristics, the Multiple Primer Analyzer online tool (Thermo Fisher Scientific) was used. DNA from *B. licheniformis* DSM13 and *B. licheniformis* B3-15 was used to validate primers and PCR conditions.

#### 2.4.2. RNA Isolation and Reverse Transcription

Aliquots (1 mL) from *B. licheniformis* B3-15 cultures in MGV or MGV-op, incubated at 45 °C, were collected at different times (12, 24, and 48 h) and centrifuged at 8000 rpm × 10 min. To obtain RNAs, the cells were treated with Trizol Reagent (Life Technologies, Carlsbad, CA, USA). To remove the traces of DNA, the samples were treated for 30 min at 37 °C with 1 U of RNase-free DNase (Promega Corporation, Madison, WI, USA). An RQ1 DNase stop solution (1 μL) was used to stop the reaction, and the sample was incubated at 65 °C for 10 min. A total of 1 μg of RNA, spectrophotometrically quantized, was used for the reverse transcription of complementary DNA (cDNA). An equal amount of RNA for each sample was reverse-transcribed into cDNA using Improm II reverse transcriptase (RT) (Promega Corporation, USA). The RT reaction was carried out at 25 °C for 5 min, then at 37 °C for 60 min and 70 °C for 15 min.

#### 2.4.3. Quantitative Real-Time Polymerase Chain Reaction (qRT-PCR)

To evaluate the gene expression of bacterial cells in the different conditions, the qRT-PCR was performed as reported previously [50] using a 7500 Fast Real-Time PCR System. The reaction was carried out with the Sso Advanced universal SYBR1 Green supermix (BioRad, Laboratories, Heracles, CA, USA), and the following operating conditions were selected: 3 min at 95 °C, followed by 35 cycles of 15 s at 95 °C and 45 s at the suitable melting temperatures for each primer, as reported in Table 2. Each sample was tested in triplicate, and data are expressed using the 2^−ΔΔCt^ (Ct) method. The differences in the expression were indicated as fold changes with respect to MGV and normalized to the levels of the elongation factor-Tu.

### 2.5. Antibiofilm Activity of BS B3-15

#### 2.5.1. Pathogenic Bacterial Strains

The strains used in this study *(Pseudomonas aeruginosa* ATCC 27853 and *Staphylococcus aureus* ATCC 29213) were acquired from the American Type Culture Collection (LGC Promochem, Teddington, UK). The *P. aeruginosa* was maintained in Luria Bertani broth (LB, Sigma Aldrich) and LB agarized with 2% Bacto agar (Difco, Baltimore, MD, USA), whereas, to maintain the *S. aureus*, Tryptic Soy Broth or Tryptic Soy Agar (TSB, TSA) was used.

#### 2.5.2. Antibiofilm Activity of BS B3-15 on Polystyrene Surfaces at Increasing Concentrations

To assess the effect of the BS B3-15 on the biofilm formation of *P. aeruginosa* and *S. aureu*s in 96-well polystyrene microplates (Thermo-Fisher Scientific, Milan, Italy), the assay previously reported by O’Toole et al. [51] was performed. Each well of the microplate (six replicates) was filled with 180 μL of *P. aeruginosa* or *S. aureus* culture (OD_600nm_ = 0.1), grown overnight in LB or TSB, respectively. BS B3-15 (20 μL) at the final concentrations of 50, 100, 200 or 300 µg/mL was added to each well, and Phosphate Buffer Saline (sPBS, Sigma Aldrich) was used as control.

#### 2.5.3. Antibiofilm Activity Assay on Polystyrene Surfaces of BS B3-15 at Different Times

BS B3-15’s ability to hinder the *P. aeruginosa* and *S. aureus* biofilm formation was evaluated as previously reported [50]. Briefly, an aliquot (20 µL) of the BS B3-15 solution (300 µg/mL) or PBS (used as control) was added to each well, containing 180 μL of each culture as reported above, at different times of bacterial growth (0, 2, 4, 8, and 24 or 48 h). The BS B3-15 was added at different times corresponding to the phases of adhesion (0), reversible attachment (2 h), irreversible attachment (4 and 8 h), and mature biofilm (48 h for *P. aeruginosa* or 24 h for *S. aureus*) of the pathogenic strains. After incubation at 37 °C for 48 h (for *P. aeruginosa*) or 24 h (for *S. aureus*) in a static condition, the cultures were gently taken from each well; to remove non-adherent bacteria, each well was washed with distilled water three times, and biofilms were stained with 0.1% (*w*/*v*) crystal violet solution and incubated at room temperature for 20 min. To eliminate the excess stain, the plates were washed with distilled water (5 times) and air-dried (for 45 min). Aliquots (200 µL) of absolute ethanol were added to each well to dissolve crystal violet from the stained biofilm. The biofilm mass was determined using the absorbance (OD_585 nm_) of the de-staining solution measured using a microtiter plate reader (Multiskan GO, Thermo Scientific, Waltham, MA, USA). The reduction in biofilm formation was expressed by applying the following formula:Reduction in biofilm formation (%) = (OD _(585 nm control)_ − OD _(585 nm sample)_)/OD _(585 nm control)_ × 100

Each data point was averaged from six replicated microwells, and the standard deviation (SD) was calculated. Statistical significance (** *p* ≤ 0.01 or * *p* ≤ 0.05) was determined using one-way ANOVA.

The multicellular structures of the biofilms onto the polystyrene surface, not treated or treated with the BS B3-15, were observed using confocal Laser Scanning Microscopy, with the TCS SP2 microscope (Leica Microsystems Heidemberg, Mannheim, Germany), equipped with Ar/Kr laser, and coupled to a microscope (Leica DMIRB). Sterile polystyrene strips (0.5 cm × 1 cm) were placed into each well of 96-well microplates containing 180 µL of overnight cultures (OD_600nm_ = 0.1) of *P. aeruginosa* in LB or *S. aureus* in TSB and BS (20 µL) at a final concentration of 300 μg/mL or sterile PBS (used as control). The microplates were incubated at 37 °C for 24 h (for *S. aureus*) or 48 h (for *P. aeruginosa*). The strips were removed from the wells and washed five times with sterile PBS to take out non-adherent bacteria; then, SYTO9 (20 µg/mL) (LIVE/DEAD Bac-light Thermo Fisher Scientific) was added to each strip to stain adherent cells, and after dark incubation (30 °C for 5 min), the biofilm on the strips was observed. The 3D images were elaborated using COMSTAT ImageJ software1.54d [52].

#### 2.5.4. BS B3-15 Antibacterial Activity

To determine the minimum inhibitory concentration (MIC) values of the BS, the serial dilution assay was carried out in microplates as previously reported [53]. Briefly, the BS B3-15 was serially diluted (1000, 500, 400, 300, 200, and 100 µg/mL) in Mueller Hinton Broth (MHB, Sigma-Aldrich, Milan, Italy) using microplates. After that, each well, containing different concentrations of BS B3-15, was inoculated with suitable aliquots of each strain suspension (OD_600nm_ = 0.1) in MHB. The microplates were placed at 37 °C overnight, and the growth of the strain was evaluated by measuring the optical absorbance (OD_600nm_). The inhibitory activity of BS B3-15 was confirmed by inoculating suitable aliquots (100 µL) from each well without visible growth onto LB or TSA plates for *P. aeruginosa* and *S. aureus*, respectively, and incubating them overnight at 37 °C.

To evaluate the antibacterial activity, the standard disk diffusion method (Kirby Bauer test) was used as reported by the National Committee for Clinical Laboratory Standard (NCCLS 2000). Suspensions (OD_600nm_ = 0.1) of each strain were prepared in 3 mL of 0.9%NaCl from an overnight culture in LA or TSA, and aliquots of each suspension (100 μL) were inoculated onto triplicate plates of Mueller Hinton agar (Sigma-Aldrich, Milan, Italy). Sterile filter paper disks (6 mm in diameter, Oxoid) loaded with 300 μg of BS B3-15 were placed onto inoculated plates and incubated overnight at 37 °C. The size of the complete inhibition zone of each disk was determined, and the mean and standard deviation (*n* = 3) were calculated.

#### 2.5.5. Bioluminescence Inhibition Assay

A bioluminescence inhibition assay was used to evaluate the potential interference of BS B3-15 in the bacterial quorum-sensing mechanism involved in the biofilm formation. Moreover, the test was used to detect the BS’s toxic effects. Briefly, 20 mL of the standard medium Sea Water Complete (tryptone 5 g/L, yeast extract 3 g/L, glycerol 3 mL/L, 750 mL/L of seawater, and 250 mL/L of distilled water) in a flask was inoculated with *Vibrio harveyi* strain G5 and incubated at 28 °C overnight. Each well of a 96-well microtiter plate was filled with aliquots (80 µL) of the *V. harveyi* overnight culture (OD_600nm_ = 0.5, equivalent to 5 × 10^8^ bacteria/mL). An aliquot (20 µL) of BS B3-15 solution, dissolved in 2%NaCl, or sterile 2%NaCl solution as control, was added to each well to reach a final BS concentration ranging from 125 to 1000 µg/mL. After 15 min of incubation at 25 °C, the luminescence of the bacterial cell suspension was evaluated and expressed as the relative luminescence unit (RLU), calculated as follows: RLU = luminescence/OD_600nm_. The Effective Concentration (EC_50_), as 50% of the RLU reduction, was used to indicate the BS’s toxicity relative to the control.

### 2.6. Antibiofilm Activity Assay on Human Nasal Epithelial Cells of BS B3-15

#### 2.6.1. Human Nasal Epithelial Cells Culture

The Human Nasal Epithelial Cells (HNEpC) were obtained from the PromoCell (Cat. Num. C-12620, Heidelberg, Germany). HNEpC was cultured in RPMI 1640 (Invitrogen Cergy Pontoise, France), complemented with penicillin/streptomycin/amphotericin (Sigma Aldrich, Saint Louis, MO, USA) and 10% Fetal Bovine Serum (Sigma Aldrich, USA origin), in a 75 cm^2^ flask and placed in the incubator at 37 °C with 5% CO_2_ [54]. To form confluent monolayers, cells from cell monolayers treated with 5 mL Trypsin EDTA (Euroclone, Milan, Italy) were resuspended at a concentration of 2.5 × 10^5^ cells/mL in culture medium and poured into each of the 96-well cell culture plates and incubated at suitable conditions.

#### 2.6.2. Antibiofilm Activity Assay on Human Nasal Epithelial Cells of BS B3-15 with Increasing Concentrations of BS B3-15

To evaluate the effect of BS B3-15 on the adhesion of *P. aeruginosa* and *S. aureus* to HNEpC, the procedure proposed by Fernandes de Oliveira [55] was used. The RPMI was removed through careful aspiration, and HNEpC was washed three times with RPMI 1640 deprived of antibiotics and serum (RPMI-1). To infect the human cells, suspensions (100 µL) in RPMI-1 of *P. aeruginosa* or *S. aureus* (1 × 10^6^ CFU/mL) were added to each well. BS, dissolved in RPMI-1 at different final concentrations (from 50 to 300 µg/mL), or RPMI-1, used as control, was added to HNEpC, and the microplates were incubated for 2 h (37 °C, 5% of CO_2_). The cells were washed with RPMI-1 (two times) and PBS (one time) to remove the non-adherent bacterial cells. After that, cold, distilled, sterile water (100 µL) was poured into each well to induce the lysis of the HNEpC, and non-adherent bacterial cells were recovered. The suspensions of non-adherent bacterial cells were serially diluted ten-fold in PBS and plated onto cetrimide agar (Oxoid) for *P. aeruginosa* or onto mannitol salt agar (Oxoid) for *S*. *aureus*. The plates were incubated at 37 °C for 18–24 h, and the Colony-Forming Units (CFUs) were counted.

#### 2.6.3. BS B3-15 Cytotoxicity

HNEpC, treated or not treated with BS B3-15 at different concentrations (100, 200, 300, 400, 500 µg/mL), was incubated at 37 °C for 24 h and 4 days, with 5% CO_2_. To evaluate the HNEpC’s viability, the cells were stained with a diluted solution (1:2000, *v*/*v*) of TO-PRO3 (Thermo Fisher Scientifics). Then, the microplates were incubated at 4 °C for 15 min in the dark and analyzed using flow cytometry (FACS Canto II).

### 2.7. Statistical Analysis

Averages and standard deviations (SDs) were calculated in all the experiments (in triplicate). Statistically significant differences among the groups were calculated using two-way ANOVA followed by Tukey’s multiple comparisons tests (** *p* ≤ 0.01, * *p* ≤ 0.05) using GraphPad Prism (version 9.0; GraphPad Software, San Diego, CA, USA).

## 3. Results

### 3.1. Optimization of Culture Conditions for BS B3-15 Production

The effects of glucose (GLU) or saccharose (SAC) at different concentrations (from 0 to 7%, *w*/*v*) in the MGV medium on the growth and biosurfactant production, estimated as the emulsifying index (E_24_) of *Bacillus licheniformis* B3-15 (BS B3-15), are shown in Table 3. The presence of GLU or SAC at 5% showed the highest values of bacterial growth (OD_600nm_ = 6) after 48 h incubation at 45 °C. However, the CFS obtained with SAC 5% was surface-active (oil drop-collapse assay) and exhibited the highest emulsifying activity (E_24_ = 60%).

The RSM-CCD, employed for multiple regression analysis to optimize the culture conditions, elucidated the effects of different concentrations of saccharose (SAC) (0–7%) and phosphate (0.06–0.60 mM) in the MGV medium on the biosurfactant production (evaluated as E_24_), which were analyzed in a set of 13 experiments designed by Minitab 21.0, and the predicted value and actual value are listed in Table 4. The emulsifying activity (E_24_) was used as a response, and the obtained interaction was fitted as a quadratic model (Equation (1)) to predict the optimized conditions via ANOVA using Minitab 21.

The derived equation can predict the E_24_ as a function of SAC (%) and phosphate concentrations, P (mM), in the ranges investigated.
E_24_ = 5.29 + 16.34 SAC + 0.071 P − 1.169 SAC × SAC (1)

The response surface plot for the parameters and selection of the best conditions for the BS production (evaluated as E_24_) as the response are shown in Figure 1.

According to ANOVA, the quadratic model of the present study was proven to be significant (*p* < 0.0001). The model’s R^2^ and F values were found to be 0.917 and 15.6, respectively, which further supports the significance of the model. Both saccharose and phosphate concentrations influenced the biosurfactant production. Overall, the predicted conditions for the highest E_24_ (71%) were saccharose 5% and phosphate (0.06 mM) (*p* ≤ 0.05).

### 3.2. BS B3-15 Production

The growth curves of *B. licheniformis* B3-15 (OD_600nm_), the emulsification activity of its CFS (E_24_%), and the crude BS B3-15-yield (mg/L) in non-optimized (MGV) and optimized (MGV-op) cultural conditions are reported in Figure 2.

The final growth (48 h) in MGV-op was lower (OD_600nm_ = 4.1) than in MGV (OD_600nm_ = 5.9); however, the emulsification activity (E_24_%) of CFS from MGV-op was often greater (Figure 2a). The production of BS B3-15 was higher in MGV-op (1.51 g/L), with an early start to production (12 h incubation), than in MGV (24 h).

### 3.3. BS B3-15 Extraction, Characterization, and Surface Activity

#### 3.3.1. BS Characterization by ATR-FTIR

To better characterize the chemical structure of the BS B3-15, ATR-FTIR analysis was performed. The wavenumbers and their assignment to the different functional groups of the crude BS are reported in Table 5. Spectra obtained using ATR-FTIR analysis, used to compare the structural features of the crude BS produced in MGV or MGV-op, overlapped, and therefore, the spectrum of BS B3-15 produced in the MGV-op condition is shown in Figure 3.

The peak observed at 3323 cm^−1^ was assigned to the OH- or NH-stretching mode. The peaks observed at 2967, 2924, and 2852 cm^−1^ and those at 1448 and 1386 cm^−1^ were attributed to the aliphatic (-CH_3_ and -CH_2_) stretching vibrations of lipids, indicating the presence of alkyl chains. The characteristics of the stretching frequencies of amides in the region 3300–3250 (Amide A) and the peaks at 1650 cm^−1^ (assigned to CO-N group) and at 1532 cm^−1^ (N-H bending of amide) were specific to surfactin-like lipopeptides.

#### 3.3.2. BS B3-15 Surface Properties

To determine the ability of BS B3-15 to modify the surface properties of hydrophobic surfaces, the contact angle assay was performed. The contact angle of water on polystyrene decreased in the presence of BS B3-15 in a dose-dependent manner. At the highest BS concentration (1600 µg/mL), the contact angle was reduced from 89° to 47.03°, indicating that BS increased the interaction between the water and the hydrophobic surface due to the reduction in surface hydrophobicity (Figure 4).

### 3.4. Expression of Lichenysin Synthetase

The surfactin gene (*srf*A) was not detected in the DNA extracted from *B. licheniformis* B3-15, nor in that extracted from its closest phylogenetically related (99.87% similarity) *B. licheniformis* ATCC 9789. The lichenysin (*lch*AA) expression by B3-15 was evaluated in MGV and MGV-op over the time of incubation (Figure 5). The gene expression was the highest after 24 h of incubation in MGV-op. However, the overexpression of the *lch*AA gene was observed after 12 h of incubation in MGV-op, rather than in the not-deprived phosphate condition (48 h).

### 3.5. Antibiofilm Activity of BS B3-15

#### 3.5.1. Addition of BS B3-15 on Polystyrene Surfaces with Increasing Concentrations

The inhibitory effect on the biofilm formation of *P. aeruginosa* and *S. aureus* of the BS is dose-dependent (Figure 6). BS B3-15 reduced the biofilm formation of *P. aeruginosa* (47.1%) more than *S. aureus* (35.9%) at the highest concentration (300 μg/mL).

#### 3.5.2. BS B3-15 Addition on Polystyrene Surfaces at Different Times from the Inoculum

The BS B3-15 (300 μg/mL) was shown to inhibit the biofilm formation at different times after the inoculum (0, 2, 4 and 8 h) and after 24 h for *S. aureus* or 48 h for *P. aeruginosa*, when the biofilm was completely established, as shown in Figure 7.

BS B3-15 added at 0 h and after 2 h strongly inhibited the adhesion and reversible attachment of *P. aeruginosa* (47.1 and 47.3% inhibition, respectively) and *S. aureus* (35.9 and 31.7% inhibition, respectively). BS B3-15 addition at 4 h and 8 h was also active with respect to the irreversible adhesion of *P. aeruginosa* (32.8 and 31.9% inhibition, respectively) and *S. aureus* (27.2 and 31.9% of inhibition, respectively). The BS was able to disrupt the preformed biofilms, being more efficient against *S. aureus* (47.2%) than *P. aeruginosa* (26.3%) (Figure 7), as also microscopically confirmed (Figure 8).

As calculated by the imaging analysis (ImageJ 1.54d), in the presence of BS B3-15, the mean thickness of the preformed biofilm of *P. aeruginosa* was reduced by 24%, and that of *S. aureus* was reduced by 45.9%.

#### 3.5.3. Antibacterial Activity

The BS (from 100 to 1000 µg/mL) did not influence the growth of *P. aeruginosa* or *S. aureus* (Appendix A). No inhibition haloes were observed using the agar diffusion assay in the presence of BS B3-15, indicating that the biopolymer did not exert any antibacterial activity at the concentrations used.

#### 3.5.4. Bioluminescent Assay

The evaluation of the possible interference of BS B3-15 on the bacterial quorum sensing and the toxicity effects were performed by bioluminescent inhibition assay [61]. The effects of BS B3-15 at different concentrations (from 0 to 1000 µg/mL) on the luminescence of *V. harveyi* G5 are reported in Appendix A. The luminescence (expressed as RLU) was constant in the presence of BS up to 500 µg/mL, and it decreased significantly in the presence of 750 µg/mL. The concentration of BS at which there was a 50% reduction in light emission was 966 ± 21.3 µg/mL (EC_50_).

### 3.6. Effects of BS B3-15 Addition to Human Nasal Epithelial Cells

The BS B3-15 (from 50 to 300 µg/mL) added to HNEpC hindered the adhesion of *P. aeruginosa* and *S. aureus* (Figure 9) after 2 h of incubation at 37 °C. BS (300 µg/mL) reduced the adhesion of *P. aeruginosa* to more than one log scale, while the *S. aureus* was reduced to five log scales (Figure 9).

The viability of HNEpC exposed to different BS B3-15 concentrations (from 50 to 500 µg/mL) after 24 h and 48 h is reported in Figure 10. The presence of crude BS at 300 µg/mL did not significantly alter the viability of HNEpC after 24 h and 48 h of exposure compared to the control. At the highest concentration (500 µg/mL), the cell viability was reduced by 33%, indicating a cytotoxic effect of BS B3-15.

## 4. Discussion

Lipopeptide surfactants (LPs), including iturin, surfactin, and lichenysins, exhibit powerful biological effects, such as antiviral and antimicrobial activities, due to their exceptional surface activity, reducing the surface and interfacial tension and the emulsifying properties [7]. LPs can promote or inhibit biofilm formation, depending upon the structure of the LP and the polarity of the cells and substrate [60]. Acting as pre-conditioning agents, LPs produced by *Bacillus* spp. were also reported to contrast with the bacterial adhesion to different surfaces, which represents a crucial step of bacterial biofilm formation due to their ability to decrease hydrophilic interactions between biotic or abiotic surfaces and bacteria [37,62]. These biosurfactant properties could be useful in controlling the microbial contamination by pathogens and their persistence as biofilms, which are a great concern in different contexts [33].

In this study, we evaluated the effects of the crude BS B3-15, produced by *B. licheniformis* B3-15, on the prevention and disruption of bacterial biofilm on polystyrene surfaces and human nasal cells. Since the biosynthesis of biosurfactants largely depends on nutritional factors [57,63,64], we first attempted to enhance the BS B3-15 yield for their large-scale production, in optimized nutritional conditions of carbohydrates, phosphates, and nitrogen, as organic (as yeast and meat extracts) and inorganic (NH_4_SO_4_) sources. We previously reported that the copresence of peptone and saccharose supported the *B. licheniformis* B3-15 growth and gave the best results in terms of sufactin-like production [44]. Although similar values of *B. licheniformis* B3-15 growth were observed in the presence of glucose and saccharose, the highest E_24_ (60%) was registered at 5% saccharose. These data are according to Mendoza et al. [65], since molasses (rich in saccharose, fructose and glucose) performed with better indices than glucose and lactose in terms of *B. subtilis* DS03 biomass and higher crude biosurfactant production. Phosphate content greatly influenced both strain growth and BS B3-15 production. As predicted by RSM analysis, the lowest phosphate concentration (0.06 mM) increased the crude BS B3-15 yield and its emulsifying activity (E_24_ = 71%). BS B3-15 production was also verified via the detection and expression of genes related to the synthesis of lichenysin and surfactin at different incubation times. Lichenysin synthetase A gene (*lch*AA) was detected using PCR in strain B3-15, but not the surfactin gene (*srf*A), nor its closest phylogenetically related *B. licheniformis* ATCC 9789, which is reported to be unable to produce surfactin and iturin-A [66,67]. After 12 h and 24 h incubation in MGV-op conditions, *lch*AA gene expression levels were two-fold greater than those of MGV, with the highest expression after 24 h. Interestingly, the early overexpression of the *lch*AA gene in MVG-op occurred during the exponential phase (12 h), with the consequent increase in the synthesis and lichenysin yield. This result suggests that, without increasing the strain biomass, the phosphate deprivation in MGV-op induced better indices of the BS-B3-15 yield (1.5 g/L), with the highest emulsified index (E_24_ = 67%) compared to MGV. The emulsifying activity of both the cell-free supernatant and crude BS-B3-15 could compete with industrially manufactured surfactants, such as Triton-X-100 (74%) and sodium dodecyl sulfate (74.4%) [44]. To better characterize the partially purified BS B3-15 obtained under optimized conditions, the FTIR spectrum was analyzed. Although lichenysin and surfactin possess a similar lipopeptide structure and also a similar pattern [68], several differences were observed as shifts in specific peaks in the most representative bands of lipopeptide components. Specifically, the spectrum of BS B3-15 showed significant shifts at lower frequency of the characteristic bands attributed to the peptide structure (amide II and amide I), suggesting that the peptide component of BS B3-15 could possess a higher molecular weight than surfactin. Moreover, the shifts of peaks in the bands attributed to aliphatic chains (C–H stretching) of fatty acids indicated that the composition of the lipidic chains of BS B3-15 was different from that of surfactin.

The crude BS B3-15 was able to hinder the adhesion and the biofilm formation on abiotic surfaces without exerting any bacteriostatic or bactericidal activity, similarly to other bacterial antibiofilm polymers, such as the exopolysaccharide mannose-rich EPS B3-15, produced by the same *B. licheniformis* B3-15 using different cultural conditions [50]. The activity of EPS B3-15 and BS B3-15 were similar in reducing the biofilm formation of *P. aeruginosa* (52.7% and 47.1%, respectively) and *S. aureus* (35.9 and 32.3%, respectively) on polystyrene surfaces. Although the emulsifying activity of EPS B3-15 was low (E_24_ = 37%) (unpublished data), its modes of action were related to the modifications of charges and the hydrophobicity of both abiotic (polystyrene and polyvinyl chloride surfaces) and bacterial surfaces, together with the modification in the expression patterns of genes involved in the early adhesion of *P. aeruginosa* and *S. aureus*. Unlike the EPS B3-15, the BS B3-15 also acted on the irreversible attachment and was able to disrupt the mature biofilms, being more efficient on *S. aureus* (47.2%) than *P. aeruginosa* (26.3%). Compared with other surfactants, such as the crude BS produced by *B. subtilis* DS03 [64] and the lipopeptide produced by *B. lichenifo*rmis AL 1.1 [32], BS B3-15 was able to similarly disrupt the preformed biofilm of *S. aureus* on the polystyrene surface, but in contrast, the BS B3-15 was also able to partially disrupt the biofilm of *P. aeruginosa*. The biofilm-disruption effects of BS B3-15 may be explained by its ability to strongly reduce the interfacial tension between the polystyrene surface and the attached cells, as assessed by the reduction in the contact angle (Figure 4) and therefore facilitating the biofilm removal. Since its addition at different concentrations did not affect the luminescence of *Vibrio harveyi* G5, BS B3-15 appears to not interfere with the quorum-sensing mechanism involved in the formation of bacterial biofilms.

In order to address its use in different medical applications, we evaluated the effects of the crude BS B3-15 on the adhesion of *P. aeruginosa* and *S. aureus* to human nasal epithelial cells (HNEpC), constituting the first barrier and one of the main targets of airborne bacterial infections. Although it is well known that human cells are sensitive to bacterial surfactants, the BS B3-15 did not show toxicity toward HNEpCs up to 300 µg/mL, probably due to its own peptide and lipidic structure. In contrast, lichenysins and surfactins from *B. licheniformis* B4094 and B4123 were reported to exert cytotoxic effects toward the tumoral Caco-2 human intestinal epithelial cells at very low concentrations (IC_50_ 16.6 and 23.5 µg/mL, respectively) [69]. BS B3-15 lichenysin (at 300 µg/mL) inhibited the adhesion of *S. aureus* (five-log scale) to human nasal cells more efficiently than *P. aeruginosa* (one-log scale). Conversely, the EPS B3-15 [50] was reported to be more effective in counteracting the adhesion of *P. aeruginosa* (five-log scale) to nasal cells than that of *S. aureus* (one-log scale). In future perspectives, the synergic action of BS B3-15 and EPS B3-15 could be evaluated to prospect their addition in a nasal spray to prevent infections of the upper respiratory tract. As assessed by the bioluminescence inhibition assay, which is also useful for detecting harmful effects of unknown substances on higher organisms in different environments [61,70], BS B3-15 could be considered safe, and therefore, its potential use as an antiadhesive could also be addressed in non-medical applications.

## 5. Conclusions

The marine, thermophilic *B. licheniformis* B3-15 represents a source of novel active biopolymers, with unique structural complexity and biocompatibility, that are useful in both environmental and human health.

The novel formulation of MGV optimized at low concentrations of phosphate induced the early production of the lichenysin-like lipopeptide, with better indices of BS-B3-15 yield and emulsified activity. Without exerting any antibacterial activity, BS B3-15 affected the early adhesion of *P. aeruginosa* and *S. aureus* on polystyrene surfaces, and it was also able to disrupt their preformed biofilms.

The BS B3-15 also inhibited the bacterial adhesion to HNEpC without interfering with cellular viability. This BS could be successfully utilized as a detergent and antiadhesive agent in industry (e.g., food, agriculture, cosmeceuticals) and for numerous medicine purposes, as nasal spray, or as a coating agent in functionalized devices (i.e., orthopedic and endotracheal devices, vascular and urinary catheters) to prevent the formation of and to remove preformed biofilms.

## Figures and Tables

**Figure 1 microorganisms-11-01842-f001:**
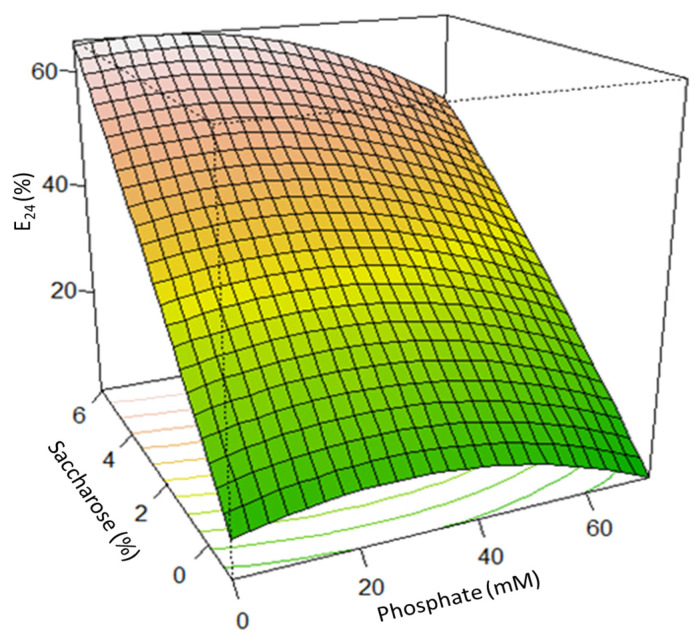
Effect of the saccharose (%) and phosphate (mM) concentrations on the emulsifying activity, expressed as E_24_ (%) of the B3-15 cell-free supernatant. The different colors on the surface plot represent the gradient range, from the lowest (green) to the greatest (light-red), for the BS production.

**Figure 2 microorganisms-11-01842-f002:**
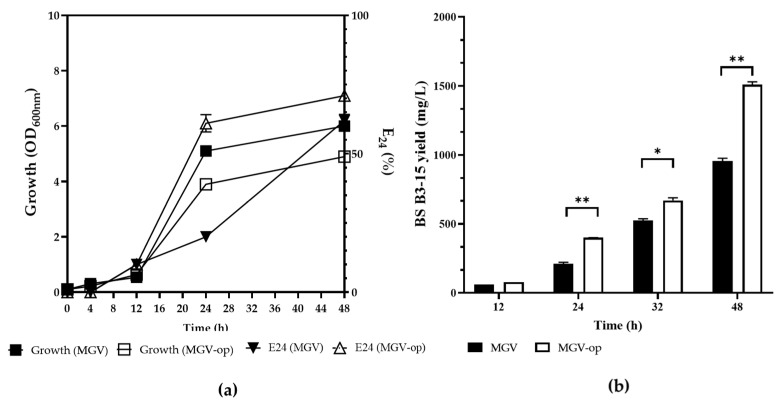
Comparison of the *B. licheniformis* B3-15 (**a**) growth (OD_600nm_) and the emulsifying index (E_24_%) of cell-free supernatants (1:1 kerosene) and (**b**) the crude biosurfactant yield (mg/L) at up to 48 h of incubation in non-optimized (MGV) and optimized media (MGV-op). Data are expressed as averages and standard deviations (*n* = 3). * Significantly different (*p* ≤ 0.05) or ** *p* ≤ 0.01.

**Figure 3 microorganisms-11-01842-f003:**
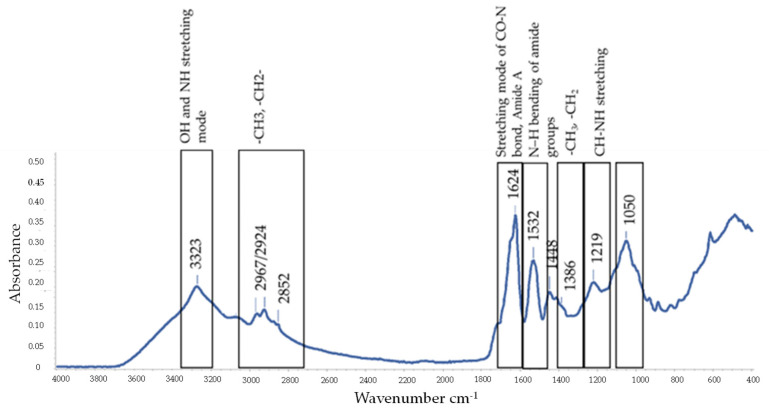
ATR-FTIR spectrum of the crude BS B3-15 produced by *B. licheniformis* B3-15 in optimized conditions (MGV-op) after 48 h incubation at 45 °C.

**Figure 4 microorganisms-11-01842-f004:**
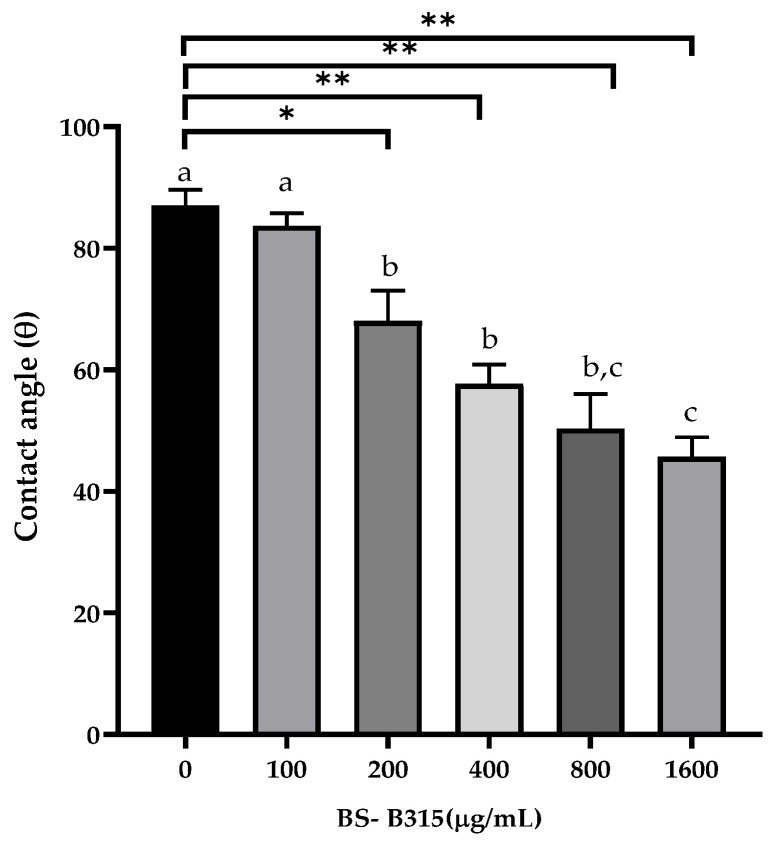
Surface properties, measured as contact angle (θ) values, with or without the BS B3-15 at different concentrations (0, 100, 200, 400, 800, and 1600 µg/mL). The bars represent the data mean ± SD for three replicates (*n* = 3). Statistical differences were evaluated using two-way ANOVA with Tukey’s multiple comparisons test. Significantly different, * *p* ≤ 0.05 and ** *p* ≤ 0.01, compared with untreated controls. Different lowercase letters above the bar graph indicate significant statistical differences (*p* < 0.01).

**Figure 5 microorganisms-11-01842-f005:**
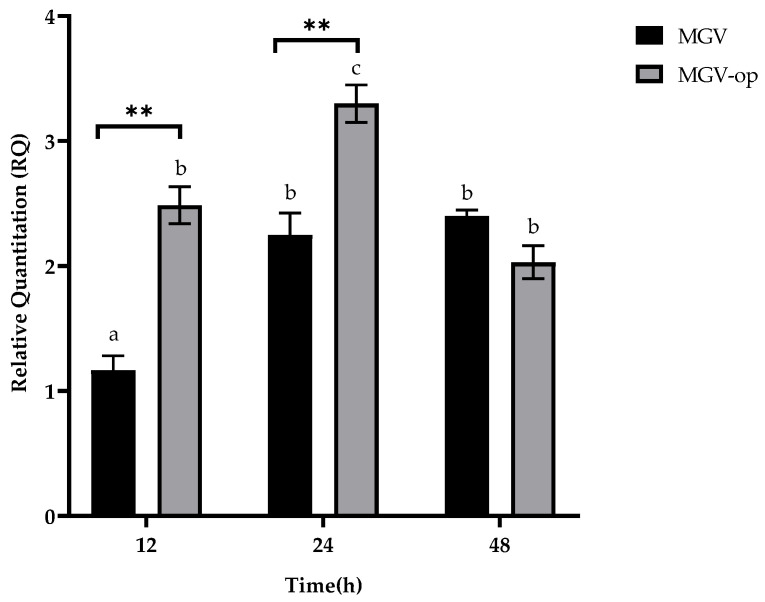
Relative quantitation (RQ) of *lch*AA at different incubation times (12, 24, and 48 h) of B3-15 in MGV or MGV-op. The bars represent mean ± SD for three replicates (*n* = 3). Statistical differences were evaluated using two-way ANOVA with Tukey’s multiple comparisons tests. Significantly different ** *p* ≤ 0.01 compared with untreated controls. Different lowercase letters above the bar graph indicate significant statistical differences (*p* < 0.01).

**Figure 6 microorganisms-11-01842-f006:**
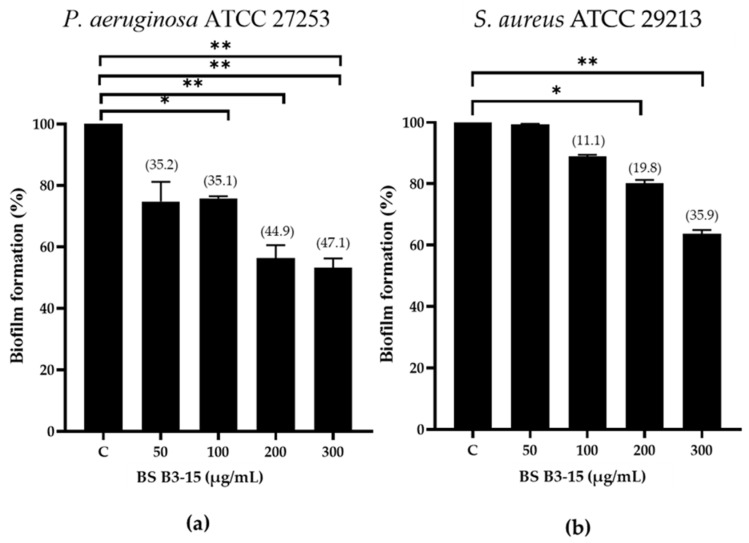
*Pseudomonas aeruginosa* ATCC 27853 (**a**) and *Staphylococcus aureus* ATCC 29213 (**b**) biofilm formation (%) on polystyrene microplate without (C) or with the BS B3-15 at different concentrations (50, 100, 200, and 300 µg/mL). The bars represent the mean ± standard deviation for six replicates (*n* = 6). * *p* ≤ 0.05 and ** *p* ≤ 0.01 show significant differences compared with untreated controls. Data on biofilm reduction (%) are reported in brackets.

**Figure 7 microorganisms-11-01842-f007:**
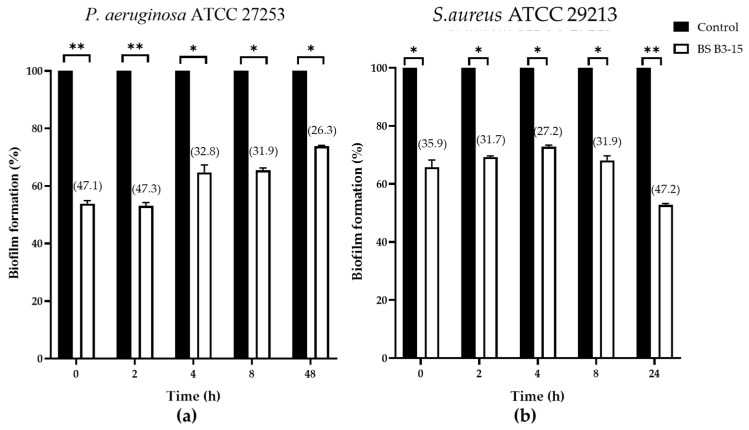
*Pseudomonas aeruginosa* ATCC 27853 (**a**) and *Staphylococcus aureus* ATCC 29213 (**b**) biofilm formation (%) on polystyrene microplate without (Control) or after the addition of the BS B3-15 (300 μg/mL) at different bacterial growth times (T0, T2, T4, T8) and after T48 for *P. aeruginosa* and T24 for *S. aureus*, when the biofilms were completely established. The bars represent mean ± SD for six replicates (*n* = 6) * *p* ≤ 0.05 and ** *p* ≤ 0.01 show significant differences compared with untreated controls. Data on biofilm reduction (%) are reported in brackets.

**Figure 8 microorganisms-11-01842-f008:**
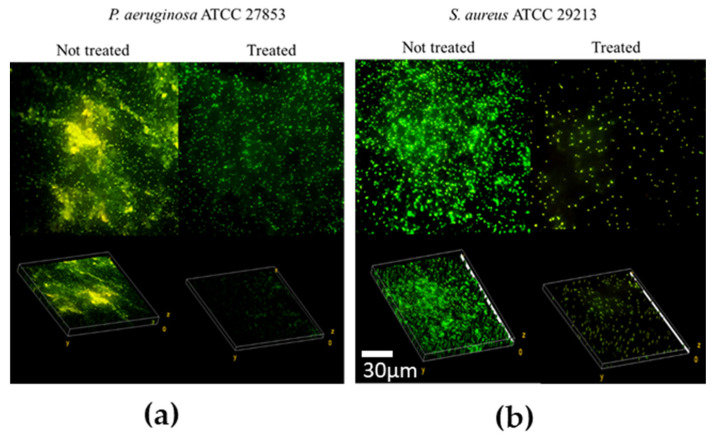
Confocal laser images (×600) of preformed biofilms from (**a**) *Pseudomonas aeruginosa* ATCC 27853 and (**b**) *Staphylococcus aureus* ATCC 29213 on polystyrene surfaces, not treated or treated with BS B3-15 (300 μg/mL).

**Figure 9 microorganisms-11-01842-f009:**
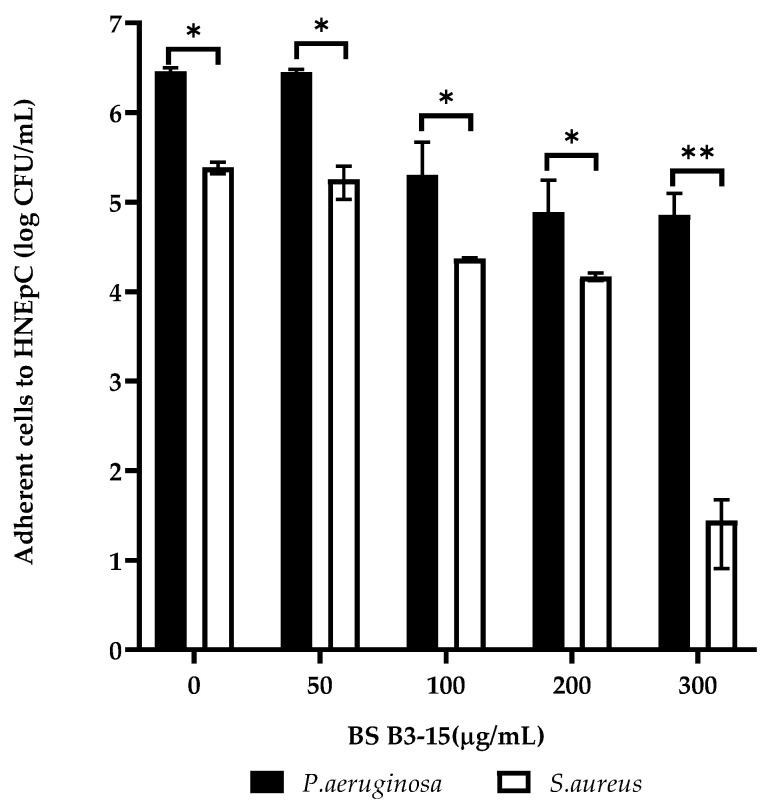
Adherence of *Pseudomonas aeruginosa* ATCC 27853 and *Staphylococcus aureus* ATCC 29213 to human nasal epithelial cells (HNEpCs), expressed on a logarithmic scale of CFU/mL, in the presence of BS B3-15, after 2 h of incubation at 37 °C. The bars represent the mean ± standard deviation for three replicates (*n* = 3). Statistical differences were evaluated using ANOVA coupled with two-way Tukey’s multiple-comparison tests. * *p* ≤ 0.05 and ** *p* ≤ 0.01: significant differences compared with untreated controls.

**Figure 10 microorganisms-11-01842-f010:**
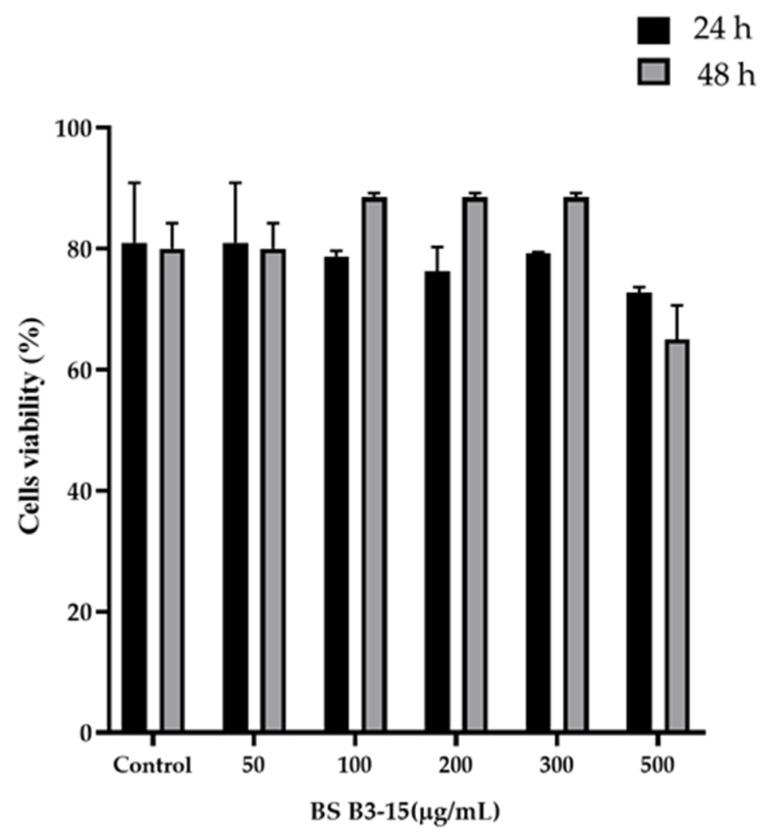
Viability of HNEpC exposed to different BS B3-15 concentrations (from 50 to 500 µg/mL) after 24 h or 48 h. Bars represent the data’s mean ± SD for six replicates (*n* = 6).

**Table 1 microorganisms-11-01842-t001:** Composition of the novel medium MGV.

Nutrient	Concentration (g/L)
NaCl	20
(NH_4_)_2_SO_4_	10
MgSO_4_ 7H_2_O	0.5
Meat Extract	3
Peptone	5
Yeast extract	3
K_2_HPO_4_	0.008–8
KH_2_PO_4_	0.002–2
Saccharose or glucose	0–70

**Table 2 microorganisms-11-01842-t002:** qRT-PCR primer sequences used in this work.

Gene	Function	Primer	Temperature Annealing (°C)	Reference
*lch*AA	Lichenysin synthetase	F: 5′-ACTGAAGCGATTCGCAAGTT-3′	56	[30]
		R: 5′–TCGCTTCATATTGTGCGTTC-3′		
*srf*A	Surfactine synthetase	F:5′- ATGAGCCAACTCTTCAAATCATTTGAT-3′	52	[48]
		R: 5′-TCACGGTTGAATGATCGGATGCTGATT-3′		
*tuf*GP	Elongation factor-Tu (housekeeping gene)	F: 5′-ACGTTGACTGCCCAGGACAC-3′	55	[49]
		R: 5′GATACCAGTTACGTCAGTTGTACGGA-3′		

**Table 3 microorganisms-11-01842-t003:** Final biomass (OD_600nm_), surface activity (oil drop-collapse assay), and emulsifying activity (E_24_) of CFS obtained from MGV in the presence of glucose (MGV + GLU) or saccharose (MGV + SAC) at different concentrations (from 0 to 7% *w*/*v*) after 48 h incubation.

Glucose or Saccharose (%)	Growth (OD_600nm_)		E_24_ (%)		Oil Drop-Collapse Assay	
	MGV + GLU	MGV + SAC	MGV + GLU	MGV + SAC	MGV + GLU	MGV + SAC
0	3.6 ± 0.1	3.2 ± 0.2	0 ± 0.3	0 ± 0.1	−	−
3	4.3 ± 0.3	4.1 ± 0.3	20 ± 1.1	5 ± 0.1	+	−
5	6.0 ± 0.2	5.9 ± 0.3	0 ± 0.1	60 ± 1.9	−	+
7	5.0 ± 0.3	5.3 ± 0.3	0 ± 0.1	30 ± 1.1	−	+

**Table 4 microorganisms-11-01842-t004:** Factorial designs via RSM-CCD with predicted and obtained responses (E_24_).

Run	x_1_	x_2_	Saccharose (%)	Phosphate (mM)	E_24_ (%) _(Predicted)_	E_24_ (%) _(Actual)_
1	0	0	2.5	30	34	41
2	0	1.414	2.5	84	28	27
3	−1	−1	0	0.06	12	15
4	0	−1.414	2.5	0.084	18	17
5	−1	1	0	60	15	14
6	1	−1	5	0.06	69	71
7	0	0	2.5	30	43	46
8	0	0	2.5	30	39	41
9	0	0	2.5	30	49	48
10	1	1	5	60	62	51
11	0	0	2.5	30	32	32
12	−1.414	0	0	30	14	15
13	1.414	0	7.0	30	52	55

**Table 5 microorganisms-11-01842-t005:** ATR-FTIR wavenumber values (cm^−1^) and band assignment to functional groups.

Wavenumber Values (cm ^−1^)	Band Assignment	References
3300–3200	OH-stretching and NH-stretching	[56,57]
3000–2800	CH_2_ and CH_3_ of lipids	[58]
1600–1700	Amide A	[59]
1690–1618	Amide I, C-O-peptidic conformation	[56,59]
1548–1530	Amide II N-O peptidic conformation	[57]
1456–1453	-CH_2_ of lipids	[58]
1400–1380	CH_2_ and CH_3_ of lipids, dipicolinic acid, amide III	[58]
~1250	CH-NH stretching	[56,57,58]
1055–1050	Phosphate groups	[56,57,58]
1035–1030	Stretching vibrations of the C–O group in esters	[60]

## Data Availability

Not applicable.

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
