# Peer review of "Lichenysin-like Polypeptide Production by Bacillus licheniformis B3-15 and Its Antiadhesive and Antibiofilm Properties"

_microorganisms, 2023, doi:10.3390/microorganisms11071842_

Round 1

Reviewer 1 Report

The publication by Vincenzo et al. focuses on the surface-active properties of a polypeptide produced by one strain of Bacillus bacteria. The paper contains many interesting results, however I have some comments that should help improve the publication:

1.       Line 33: „(1 log and 5 log, respectively)” – I suggest to use more precise “(10 x and 10,000 x higher, respectively)”. The “log” should be followed by some value or symbol.

2.       Lines 39-47: Couldn’t biosurfactants be considered as bioemulsifiers?

3.       Table 3: Standard deviation or standard error are missing.

4.       Line 358: “SAC” abbreviation should be explained.

5.       Figure 1: The units should be added to axis labels.

6.       Table 5, Fig. 3, and their descriptions: “Amide A” or “Amide I/II” are not precise. Please provide the exact references (i.e. which signal is from O-H bonds or N-H etc.).

Author Response

 Line 33: „(1 log and 5 log, respectively)” – I suggest to use more precise “(10 x and 10,000 x higher, respectively)”. The “log” should be followed by some value or symbol.

R: The text has been modified as suggested.

  1. Lines 39-47: Couldn’t biosurfactants be considered as bioemulsifiers?

R: Not all bio emulsifier could be considered as biosurfactants. The terms biosurfactant and bioemulsifier have often been used interchangeably to describe surface active biomolecules. However, it is important to note that there are marked differences between them especially based on their physico-chemical properties, molecular weight and consequently their roles in nature and biotechnological applications (Perfumo et al., 2010; Smyth et al, 2010). 

  1. Table 3: Standard deviation or standard error are missing.

R: The standard deviation has been added.

  1. Line 358: “SAC” abbreviation should be explained.

R:The SAC abbreviation has been explained in the text.  

  1. Figure 1: The units should be added to axis labels.

R: The units have been added in the Figure 1.

  1. Table 5, Fig. 3, and their descriptions: “Amide A” or “Amide I/II” are not precise. Please provide the exact references (i.e. which signal is from O-H bonds or N-H etc.).

R: Table 5 and Figure 3 have been corrected. In details, the FTIR band ranging from 1600 to 1700 cm-1 was assigned to Amide A, and the band 3300-3200 cm- was assigned to OH-stretching and NH-stretching.

Reviewer 2 Report

In this article, Zammuto and co-workers described the anti-adhesive and anti-biofilm activity of the biosurfactant BS B3-15 obtained from the free-cell supernatant of Bacillus licheniformis B3-15. Applying the Response Surface Methodology (RSM-CCD), they optimized the growth medium to enhance the production yield of the bioactive compound. Furthermore, they characterized the structural and functional properties of BS B3-15 and demonstrated its capacity to inhibit biofilm formation of Pseudomonas aeruginosa ATCC 27853 and Staphylococcus aureus ATCC 29213 on polystyrene surfaces. The biosurfactant proved to be active in different stages of the biofilm formation process, disrupting also the mature biofilms with different efficacy against the two pathogenic strains. Finally, the author demonstrated that BS B3-15 reduced the adhesion of P. aeruginosa and S. aureus to human Nasal Epithelial cells without cytotoxic effects. The article presents very interesting data, suggesting the potential application of the biosurfactant BS B3-15 in medical fields.

However, I suggest modifying the “Results” section to better introduce the aim and the procedure of each experiment. Often such indications are reported in the “Materials and Method” section. Hence, also this section should be revised to explain just the methodological aspect of the experiments. As an example, lines 123-128 should be located at the beginning of paragraph 3.1. These corrections are crucial to make the text comprehensible for readers.

Furthermore, the punctuation should be corrected throughout the text.

Other specific comments are here listed:

-          Line 88: change “biofilm” with “biofilm formation”

-          Line 97: change “yield” with “production yield”

-          Line 100 and elsewhere: “incubation” is referred to bacterial growth. Please clarify this point

-          Lines 104-106: the period is not complete

-          Line 113 and elsewhere: change 25°C with 25 °C to keep consistency throughout the manuscript

-          Line 208-209: please clarify the sentence “45 s. the Table 2 reported the annealing temperatures for each primer”

-          Lines 222, 230, 300 and 310: change the titles to better describe the paragraph content. As an example, line 222 may be written in this way “Antibiofilm activity assay on polystyrene surfaces with increasing concentrations of BS B3-15”

-          Lines 222 and elsewhere: change “BS” with “BS B3-15” to keep consistency throughout the manuscript

-          Line 233, 438, 444 and elsewhere: “at different times” refers to the time from the inoculum. Please clarify this point

-          Line 270: please explain MHB meaning

-          Line 279: delete “ . and.”

-          Line 318: delete the bracket after “The cell”

-          Line 345: change “Growth (OD600nm), surface active” with “Final biomass (OD600nm), surface activity”

-          Line 364-365: please clarify the caption

-          Line 411 and elsewhere: change the lowercase letters with asterisks to keep consistency throughout the manuscript

-          Line 430: correct the punctuation

-          Line 444: delete “(T24)”

-          Line 453: change “Figure 9” with “Figure 7” and explain better the confocal Laser Scanning Microscopy experiment. Do you evaluate biomasses, average thicknesses, and roughness coefficient of treated and untreated biofilms? The caption of Figure 8 (line 455) should also be improved

-          Lines 463-468: in my opinion, it is not very clear the contribution of this experiment. Do you evaluate a potential interference of quorum sensing? How does this assay give information about the toxicity of BS B3-15? Maybe this point must be argued also in the discussion section.

-          Lines 477 and 478: “The bars represent the mean ± SD for three replicates (n = 3)” is written twice

-          Lines 499 and 500: I suggest developing some points regarding the connection between the microbial biofilm and biosurfactants

-          Line 507: change “Mendoza et [65]” with “Mendoza et al. [65]”

-          Lines 455 and 456: it is not clear in the sentence that “biofilm developed on polystyrene and polyvinyl chloride surfaces” refers to mature/pre-formed biofilm. Please specify this point.

- Lines 552 and 555: “The disruption effects of BS B3-15 may be explained by its ability to strongly reduce the interfacial tension between the polystyrene surface and the attached cells, and therefore facilitating the biofilm remove”. Please indicate the results assuring that (oil drop-collapse assay? Figure 4?). 

Moderate editing of English language is required.

The punctuation should be corrected throughout the text.

Author Response

I suggest modifying the “Results” section to better introduce the aim and the procedure of each experiment. Often such indications are reported in the “Materials and Method” section. Hence, also this section should be revised to explain just the methodological aspect of the experiments. As an example, lines 123-128 should be located at the beginning of paragraph 3.1. These corrections are crucial to make the text comprehensible for readers.

R: Results section has been modified accordingly.

Furthermore, the punctuation should be corrected throughout the text.

Other specific comments are here listed:

-          Line 88: change “biofilm” with “biofilm formation”

R: Done

-          Line 97: change “yield” with “production yield”

R: Done

-          Line 100 and elsewhere: “incubation” is referred to bacterial growth. Please clarify this point

R: The sentence has been modified.

-          Lines 104-106: the period is not complete

R: The period has been modified.

-          Line 113 and elsewhere: change 25°C with 25 °C to keep consistency throughout the manuscript

R: Suggested changes have been done.

-          Line 208-209: please clarify the sentence “45 s. the Table 2 reported the annealing temperatures for each primer”

R: The sentence has been modified.

-          Lines 222, 230, 300 and 310: change the titles to better describe the paragraph content. As an example, line 222 may be written in this way “Antibiofilm activity assay on polystyrene surfaces with increasing concentrations of BS B3-15”

R: Titles have been modified as suggested.

-          Lines 222 and elsewhere: change “BS” with “BS B3-15” to keep consistency throughout the manuscript

R: Substitutions were done

-          Line 233, 438, 444 and elsewhere: “at different times” refers to the time from the inoculum. Please clarify this point

R: The sentence has been clarified

-          Line 270: please explain MHB meaning

R: The MHB acronym has been explained

-          Line 279: delete “ . and.”

Done

-          Line 318: delete the bracket after “The cell”

Done

-          Line 345: change “Growth (OD600nm), surface active” with “Final biomass (OD600nm), surface activity”

Done

-          Line 364-365: please clarify the caption

R:The caption has been modified

-          Line 411 and elsewhere: change the lowercase letters with asterisks to keep consistency throughout the manuscript

R:The figures have been arranged according your suggestions.

-          Line 430: correct the punctuation

R:The punctuation has been corrected

-          Line 444: delete “(T24)”

R: The sentence has been modified.

-          Line 453: change “Figure 9” with “Figure 7” and explain better the confocal Laser Scanning Microscopy experiment. Do you evaluate biomasses, average thicknesses, and roughness coefficient of treated and untreated biofilms? The caption of Figure 8 (line 455) should also be improved.

R: The caption of figure 8 has been modified. Please, consider that Figure 9 cannot replace figure 7, since the results are related to different substrata (i.e. in figs 7 polystyrene, and in figure 9 on human nasal cells).

-          Lines 463-468: in my opinion, it is not very clear the contribution of this experiment. Do you evaluate a potential interference of quorum sensing? How does this assay give information about the toxicity of BS B3-15? Maybe this point must be argued also in the discussion section.

R:The bioluminescent assay was performed to evaluate the potential interference of BS B3-15 in the bacterial quorum sensing and also to determine its toxic effects. Results have been discussed.

-          Lines 477 and 478: “The bars represent the mean ± SD for three replicates (n = 3)” is written twice

R: The caption has been corrected.

-          Lines 499 and 500: I suggest developing some points regarding the connection between the microbial biofilm and biosurfactants

R: The discussion section has been improved.

-          Line 507: change “Mendoza et [65]” with “Mendoza et al. [65]”

Done.

-          Lines 455 and 456: it is not clear in the sentence that “biofilm developed on polystyrene and polyvinyl chloride surfaces” refers to mature/pre-formed biofilm. Please specify this point.

R: The point has been clarified

- Lines 552 and 555: “The disruption effects of BS B3-15 may be explained by its ability to strongly reduce the interfacial tension between the polystyrene surface and the attached cells, and therefore facilitating the biofilm remove”. Please indicate the results assuring that (oil drop-collapse assay? Figure 4?). 

R: The sentence has been modified.